# MicroRNAs in Cancer and Cardiovascular Disease

**DOI:** 10.3390/cells11223551

**Published:** 2022-11-10

**Authors:** Mirolyuba Ilieva, Riccardo Panella, Shizuka Uchida

**Affiliations:** Center for RNA Medicine, Department of Clinical Medicine, Aalborg University, DK-2450 Copenhagen SV, Denmark

**Keywords:** cancer, cardiovascular disease, miRNA

## Abstract

Although cardiac tumor formation is rare, accumulating evidence suggests that the two leading causes of deaths, cancers, and cardiovascular diseases are similar in terms of pathogenesis, including angiogenesis, immune responses, and fibrosis. These similarities have led to the creation of new exciting field of study called cardio-oncology. Here, we review the similarities between cancer and cardiovascular disease from the perspective of microRNAs (miRNAs). As miRNAs are well-known regulators of translation by binding to the 3′-untranslated regions (UTRs) of messenger RNAs (mRNAs), we carefully dissect how a specific set of miRNAs are both oncomiRs (miRNAs in cancer) and myomiRs (muscle-related miRNAs). Furthermore, from the standpoint of similar pathogenesis, miRNAs categories related to the similar pathogenesis are discussed; namely, angiomiRs, Immune-miRs, and fibromiRs.

## 1. Introduction

Cancer and cardiovascular disease are the leading causes of death across the globe accounting for one in six deaths [1] and 32% of all deaths worldwide [2], respectively, according to World Health Organization (WHO). Both cancer and cardiovascular disease are the umbrella terms commonly used to describe several disease etiologies. Each etiology of cancer and cardiovascular disease (e.g., lung cancer and ischemic heart disease, respectively) has its own distinct cause and progression pattern. However, recent research suggests that many aspects of cancer and cardiovascular disease are similar in terms of pathogenesis [3,4,5], leading to the development of specific field of study called cardio-oncology [6,7]. For example, both diseases involve dysregulated functionalities in vasculature, where abnormal vasculature (called, tumor vasculature [8]) occurs in cancer, while coronary artery disease is a type of cardiovascular disease caused by the narrowing or blockage of coronary arteries [9]. Another example is the involvement of immune responses, where prolonged or chronic inflammation is a hallmark of cancer [10,11,12] as well as cardiovascular disease [13,14,15]. The activation of immune responses often leads to the deposition of excessive extracellular matrices [16,17], which are another hallmark of cancer [17] and cardiac fibrosis as the end-stage of heart failure [18].

MicroRNAs (miRNAs) are evolutionary-conserved, regulatory short [~22 nucleotides (nt)] non-protein-coding RNAs that function as translational inhibitors by binding to the 3′-untranslated regions (3′-UTRs) of messenger RNAs (mRNAs) [19,20]. As one miRNA is predicted to bind hundreds of mRNAs due to its very short seed sequence (~6 nt) [21,22,23], it is speculated and experimentally shown for some miRNAs to regulate cascades of signaling pathways and their downstream targets. Due to their versatilities, dysregulation in miRNAs is linked to a variety of diseases, including cancer [24,25] and cardiovascular disease [26,27,28]. As the regulatory importance of miRNAs is experimentally proven, the therapeutic silencing of miRNAs is being explored [29,30,31,32,33,34]. However, due to their biodistributions (e.g., including their presence in the circulation [35,36,37]) and the presence of many target mRNAs for one miRNA, the precise mechanistic elucidation of each miRNA is urgently needed to advance into clinics. Since a specific miRNA is highly dependent on which target mRNAs are present in a specific biological context, it must be taken into consideration that the same miRNA can yield different biological outcomes depending on the specific cell or tissue [38]. This is especially important when considering miRNAs as potential therapeutic targets.

As cancer and cardiovascular disease share several aspects of disease causes and progressions, it is no surprise that many miRNAs are shown to be involved in pathogeneses of both cancer and cardiovascular disease. Because the heart is the least likely organ to harbor tumor growth [39], the communication between researchers working in miRNAs for either cancer or cardiovascular biology is scarce, although many miRNAs are found to be dysregulated in both diseases. To fill this gap in knowledge, here, we summarize the current status of miRNA research from the perspective of shared disease progression mechanisms in cancer and cardiovascular disease.

## 2. OncomiRs vs. MyomiRs

According to the latest annotation provided by the GENCODE consortium (Release 41; https://www.gencodegenes.org/human/stats_41.html; accessed on 3 October 2022), there are 1879 human miRNAs. Due to the intensive miRNA research in the last three decades [40], many (but not all) miRNAs have been studied functionally and some mechanistically. To date, miRNAs have been categorized based on their functionalities. These categories include oncomiRs and (cardiac) myomiRs to describe cancer- and striated muscle-related miRNAs, respectively. Although the heart consists of cell types other than cardiac muscle (cardiomyocytes), for simplicity, here, we will compare oncomiRs and myomiRs to understand the possible overlaps of the functional miRNAs in both cancer and cardiovascular disease.

As there are many different types of cancer, the list of oncomiRs is growing rapidly due to the availability of next generation sequencing (i.e., small RNA sequencing) to identify miRNAs overexpressed in tumor samples. As such, there are several dedicated databases for oncomiRs available, including miRCancer [41], OncomiR [42,43,44], and the OncoMir Cancer Database (OMCD) [45]. Compared to oncomiRs, the list of myomiRs is small, including *miR-1*, *miR-133a/b*, *miR-206*, *miR-208a/b*, *miR-302*, *miR-367*, *miR-486*, and *miR-499* [46,47]. Not surprisingly, all myomiRs are involved in tumorigenesis.

One of the most abundant miRNAs in the heart [48], *miR-133*, is a regulator of cardiac hypertrophy [49] and its down-regulation was observed in patients with myocardial infarction [50] (Figure 1). In gastric cancer, *miR-133* is down-regulated in gastric cancer patients and negatively associated with tumor size, invasion depth, and peripheral organ metastasis [51]. Mechanistically, *miR-133* targets 3′-UTR of the cell division cycle 42 (*CDC42*) gene to regulate the downstream effectors of CDC42, P21-activated kinases (PAKs). Similarly, an overexpression of *miR-133a* in the lung cancer cell lines, A549 and NCI-H1299, results in the suppression of cell proliferation, migration, and invasion by targeting matrix metallopeptidase 14 (*MMP14*) [52]. Another study shows that the overexpression of *miR-133b* in the lung cancer cell line, A549, re-sensitized the radioresistant A549 cells by targeting pyruvate kinase M1/2 (*PKM*, also known as *PKM2*) to regulate glycolysis [53]. Besides gastric and lung cancers, functions of *miR-133* are also reported in glioblastoma [54], oral cancer [55], and prostate cancer [56]. All other miRNAs are also shown to be functionally important for tumorigenesis, suggesting the importance of examining miRNAs in onco-cardiology.

## 3. Angiogenesis: AngiomiRs

Angiogenesis is the process of new blood vessel formation through the migration, growth, and differentiation of endothelial cells [57,58]. In cancer, angiogenesis allows for a tumor to grow as new vessels provide nutrients and oxygen to malignant cells [59,60]. In cardiovascular disease, therapeutic angiogenesis aims to provide the blood flow to the ischemic heart tissue [61,62]. Thus, in both diseases, angiogenesis is an important therapeutic target, although the opposite effects are observed. During angiogenesis, several miRNAs are functionally involved, which has created a specific term to describe these angiogenesis-related miRNAs called, angiomiRs (Figure 2). AngiomiRs include *miR-15/16*, *miR-17~92* cluster, *miR-18a*, *miR-19*, *miR-21*, *miR-23b*, *miR-27a/b*, *miR-29b*, *miR-30*, *miR-34a*, *miR-57*, *miR-125b*, *miR-126*, *miR-128*, *miR-143*, *miR-145*, *miR-155*, *miR-192*, *miR-194*, *miR-199a*, *miR-200* family, *miR-204*, *miR-210*, *miR-217*, *miR-296*, *miR-378*, *miR-484*, *miR-494*, *miR-497*, *miR-542-3p*, *miR-573*, *miR-642*, and *let-7b* [63,64], which some are discussed below.

The *miR-17~92* cluster was first reported in tumorigenesis [65] and is one of the most well-studied miRNA clusters [66,67]. By crossing *miR-17~92* floxed mice with an inducible vascular endothelial cell specific Cre driver (Cdh5-cre/ERT2), Chamorro-Jorganes et al. demonstrated that retinal angiogenesis was reduced during the development of these mice [68]. Furthermore, the vascular endothelial growth factor (VEGF)-induced ear and tumor angiogenesis were reduced, suggesting that VEGF regulates *miR-17~92* cluster expression leading to the regulation of angiogenesis. The involvement of the *miR-17~92* cluster is well documented in various diseases, including cardiovascular disease [69,70]. Since the *miR-17~92* cluster consists of *miR-17*, *miR-18a*, *miR-19a*, *miR-20a*, *miR-19b-1*, and *miR-92a-1*, each miRNA in this cluster is also shown to be important for angiogenesis, including tumorigenesis and cardiovascular disease. For example, *miR-92a* is dysregulated in many forms of cancer, suggesting it is a potential diagnostic biomarker as well as a therapeutic target [71]. In the cardiovascular system, Bonauer et al. demonstrated that overexpression of *miR-92a* in endothelial cells inhibited angiogenesis in murine models of limb ischemia and myocardial infarction, while the silencing of *miR-92a* via antagomiR resulted in enhanced angiogensis and the functional recovery of the damaged tissues in murine disease models, suggesting *miR-92a* as a potential therapeutic target for ischemia diseases [72].

The *miR-200* family is another well studied miRNA family that includes *miR-141, miR-200a, miR-200b, miR-200c,* and *miR-429* [73]. In cancer, the *miR-200* family is shown to play functional roles in cell malignant transformation and preventing tumor initiation [74]. By profiling epicardial adipose tissue from coronary artery disease (CAD) patients and non-CAD atherosclerotic patients, Zhang et al. demonstrated that the expressions of *miR-141-3p*, *miR-200b*, *miR-200c-3p*, and *miR-429* are up-regulated in CAD patients compared to non-CAD patients [75]. By performing a series of experiments in vitro, the authors demonstrated that the overexpression of *miR-200b-3p* in human umbilical vein endothelial cells (HUVECs) resulted in increased apoptosis under oxidative stress. Mechanistically, *miR-300b-3p* targets histone deacetylase 4 (HDAC4) as the overexpression of HDAC4 reduced the increased apoptosis induced by inhibiting *miR-200b-3p*, suggesting that *miR-200b-3p* is a potential therapeutic target for atherosclerosis.

*MiR-34a* is a tumor suppressor and considered as a diagnostic and prognostic biomarker as well as a therapeutic target in various cancers, including head and neck squamous cell carcinoma, thyroid cancer, and cancer stem cells [76,77]. Interestingly, the expression of *miR-34a* is increased in senescent HUVECs and in the heart and spleen of older mice [78]. When overexpressed, *miR-34a* suppressed cell cycle and proliferation by inhibiting sirtuin 1 (*SIRT1*). Because ageing is a hot topic to be investigated, subsequent research shows the functional importance of *miR-34a* in cell types other than endothelial cells in the heart, including in cardiomyocytes [79,80], fibroblasts [81], and smooth muscle cells [82,83]. This is not an isolated case as many other angiomiRs (and other miRNAs) are expressed rather ubiquitously, suggesting that examining miRNAs as a common mechanism of action for cardio-oncology is not a big surprise.

## 4. Immune Responses: Immuno-miRs

Prolonged inflammation is a hallmark of cancer [10] that immune systems can have both positive and negative effects on regarding the development of tumors and prognostics of cancer patients [84]. Indeed, immunotherapy is a type of treatment using one’s own immune system to fight cancer, but the success rates of immunotherapy drugs vary between 15–30% in most tumor types, while 50–80% in melanoma [85]. As the immune system is a complex system involving many different cell types (e.g., basophils, eosinophils, lymphocytes, macrophages, monocytes, and neutrophils) to fight against infection [86,87], understanding the immune system is also important in cardiovascular disease [88,89,90]. For example, myocardial infarction leads to the loss of cardiomyocytes, which are replaced by non-contracting scar tissue [91,92]. The immune system is a double-edged sword in the remodeling process of the infarcted heart as macrophages are necessary for repair in the acute phase as their systemic depletion results in impaired scar formation and the rupture of the left ventricle of the heart. However, the accumulation of macrophages in non-infarcted regions of the left ventricle leads to progressive myocyte attrition, collagen deposition, and loss of the pump function of the heart in a chronic phase of remodeling of the infarcted heart. As miRNAs are expressed in many immune cells and finetune the important signaling pathways, the list of immuno-miRs is growing rapidly [93,94]. As such, specialized databases for immune-miRs are available, including IRNdb [95], RNA2Immune [96], and RNAimmuno [97]. In the following, examples of immune-miRs are explained in cancer and cardiovascular disease.

Monocytes are a type of white blood cells (leukocytes) that can differentiate into macrophages and dendritic cells [98]. Furthermore, monocyte-derived macrophages can be polarized into inflammatory subtype, M1, and anti-inflammatory subtype, M2 [99]. The cascade of differentiation and polarizations are controlled by the coordinate actions of cytokines, which can be regulated at the transcriptional and post-transcriptional levels, where miRNAs can regulate the translation of transcription factors responsible for cytokine gene expressions. These microRNAs include *miR-125a-3p* and *miR-26a-2* in M1 macrophages, while *miR-27a*, *miR-29b-1*, *miR-132*, *miR-193b*, and *miR-222* constitute the M2 macrophages [100] (Figure 3A). For example, *miR-222* targets ADAM metallopeptidase domain 17 (*ADAM17*) to modulate multidrug resistance in colorectal carcinoma [101] (Figure 3B). In breast cancer, the overexpression of *miR-222* inhibits the chemotaxis of tumor-associated macrophages by targeting C-X-C motif chemokine ligand 12 (*CXCL12*) [102]. In the serum, the level of *miR-222* is independently associated with atrial fibrillation (irregular heart rhythm) in patients with degenerative valvular heart disease [103]. In addition, the level of *miR-222* is elevated in acute viral myocarditis caused by Coxsackievirus B3 [104]. Although the functions of *miR-222* are mainly reported in cardiomyocytes [104,105,106] and cardiac fibroblasts [107,108], it is clear that immune-miRs in monocytes and macrophages are important regulators of immune responses in cancer and cardiovascular disease.

Enriched in immune cells, *miR-155* is a master regulator of immune responses [109] (Figure 3C). In breast cancer, the increased expression of *miR-155* is associated with high tumor grade, advanced stage, and lymph node metastasis [110]. Similarly, *miR-155* is overexpressed in other forms of cancer, including esophageal cancer [111], liver cancer [112], and lung cancer [113], which calls for *miR-155* as a diagnostic and prognostic cancer biomarker [114] as well as therapeutic target [115]. As shown in the previous section, *miR-155* is an angiomiR so its function is well known in the endothelial cells and atherosclerosis [116,117]. Besides endothelial cells, *miR-155* is highly expressed in monocytes and macrophages, which Pankratz et al. used in knockout mice to elegantly demonstrate that *miR-155* regulates angiogenesis and arteriogenesis by controlling their target genes, angiotensin II receptor type 1 (*AGTR1*) and suppressor of cytokine signaling 1 (*SOCS1*) in endothelial and monocyte/macrophages, respectively [118].

Extracellular RNAs (exRNAs) are a type of cell–cell communication that are produced by a donor cell and are released into the extracellular environment (e.g., body fluid, circulation) [119]. They are contained in the lipid particles, such as extracellular vehicles (EVs), including exosomes. ExRNAs include proteins and RNAs, including miRNAs. For example, *miR-146a-5p* is enriched in the hepatocellular carcinoma-derived exosomes [120] (Figure 3D). The transcription factor, spalt like transcription factor 4 (SALL4), binds to the promoter of *miR-146a-5p* to directly control its expression in exosomes, thereby regulates the polarization of macrophages into M2 tumor-associated macrophages. In contrast, cardiomyocyte-derived exosomal *miR-146a-5p* promotes M1 macrophage polarization while inhibiting M2 macrophage polarization by targeting TNF receptor associated factor 6 (*TRAF6*) [121]. This is just of many miRNAs contained in exosomes.

The studying of immunology has intensified in recent years due to the rise of coronavirus disease 2019 (COVID-19) [122,123,124,125]. As there is a substantial risk of heart problems associated with COVID-19 and mRNA vaccines [126,127,128,129], it is likely that more and more miRNAs will be identified in the heart, which may have been studied in cancer previously to expand the list of immune-miRs in the cardio-oncology field.

## 5. Fibrosis: fibromiRs

Fibrosis is a process in which fibroblasts and other mesenchymal cells are activated to become myofibroblasts to secrete an excess number of extracellular matrices (ECM; e.g., collagens, glycosaminoglycans, and glycoproteins) [130] (Figure 4A). It is the end stage of many diseases, including cardiovascular disease [18]. In cancer, cancer-associated fibroblasts (CAFs) promote tumorigenic features, including ECM deposition, epithelial-to-mesenchymal transition (EMT), and metastasis [131]. To understand fibrosis, many screening studies have been performed to identify differentially expressed genes and miRNAs, which are collectively called fibromiRs [132,133,134,135]. For example, *miR-21* is the most studied fibromiR [136,137,138,139] (Figure 4B). Not only is it highly expressed in many forms of cancer and suggested as potential diagnostic biomarkers of cancer types (breast, pancreatic, colorectal, and prostate cancer) [140], *miR-21* stimulates MAP kinase signaling in cardiac fibroblasts, thereby contributing to myocardial disease [141]. Furthermore, *miR-21* targets matrix metallopeptidase 2 (*Mmp2*) in cardiac fibroblasts of the infarcted heart via phosphatase and the tensin homolog (PTEN) pathway [142], suggesting the important signaling roles of *miR-21* in both cancer and cardiovascular disease.

Multiple reports show that another oncomiR, *miR-22*, is highly involved in tumor progression in multiple tumors, including breast cancer [143], acute myeloid leukemia (AML) [144], and hepatocellular carcinoma (HCC) [145,146]. The effect of *miR-22* on HCC seems to be related to the early effect of *miR-22* on liver fibrosis through its regulation of bone morphogenic protein 7 (*BMP7*) [147] (Figure 4C), which starts from a degenerative process and ultimately leads to HCC developing. Interestingly, *miR-22* is reported to have the same effect on cardiac fibrosis [148] via the regulation of *Sirt1* and *HDAC4*. As the role of *miR-22* in fibrosis is conserved in multiple diseases and tissues, this miRNA could serve as a potential therapeutic target in liver [149] and cardiac fibrosis [150].

Although fibroblasts can be found throughout the human body, they are heterogeneous populations of cells without any single cell surface marker that is specific for fibroblasts as many markers are expressed in other cell types, including epithelial and immune cells [151,152]. In this regard, microRNAs are involved in activating fibroblasts, which contribute to the heterogeneity of fibroblasts [153]. For example, the members of the *miR-200* family, *miR-141* and *miR-200a*, target C-X-C motif chemokine ligand 12 (*CXCL12*; also known as *CXCL12β*) to regulate the immunosuppressive activity of a subtype of carcinoma-associated fibroblasts in ovarian cancer [154]. Besides the *miR-200* family being angiomiRs as written in above subsection, *miR-200b* is negatively regulated by the epigenetic factor, DNA methyltransferase 3 alpha (Dnmt3a), to control autophagy in rat cardiac fibroblasts [155]. In addition, small RNA-seq experiment using rat cardiac fibroblasts induced with transforming growth factor-β1 (TGF-β1) showed 3 up- (*miR-325-3p, miR-325-5p,* and *miR-210-5p*) and 21 down-regulated miRNAs (e.g., *miR-19a-3p, miR-19b-3p, miR-144-3p,* and *miR-200b-3p*), potentially targeting genes involved in calcium and glutamatergic synapse signaling pathways [156]. Taken together, there are many shared fibromiRs between CAFs and cardiac fibroblasts.

## 6. Conclusions

To maintain the homeostasis of the tissues and remodeling of the tissues upon damages, angiogenesis, immune responses, and fibrosis are interconnected. As such, miRNAs are identified to be involved in each cellular activity as angiomiRs, immuno-miRs, and fibromiRs, respectively. Not surprisingly, some miRNAs (e.g., *miR-17~92* cluster, *miR-34a*, and *miR-200* family) are involved in all three cellular activities, which some overlapping miRNAs are responsible for such cellular activities and responses. This is particularly interesting as cancer is considered as a complex adaptive ecosystem [157], in which cancer cells and the stromal cells transform, cooperate, and even co-evolve with each other over time and space [158]. Thus, it will be interesting to further investigate miRNAs from the perspective of the ecosystem in cancer and possibly in cardiovascular disease.

As cancer and cardiovascular disease are two of the leading causes of death worldwide, it will be exciting to find a common disease mechanism. As reviewed above, miRNAs are shared between these life-threatening diseases. Given that miRNAs are investigated as potential therapeutic targets, increased communication between researchers working with cancer and cardiovascular disease is necessary to find a potential cure for these diseases. To this end, the rise of the cardio-oncology field should facilitate a further understanding of the pathogeneses of these two diseases, possibly through miRNAs.

## Figures and Tables

**Figure 1 cells-11-03551-f001:**
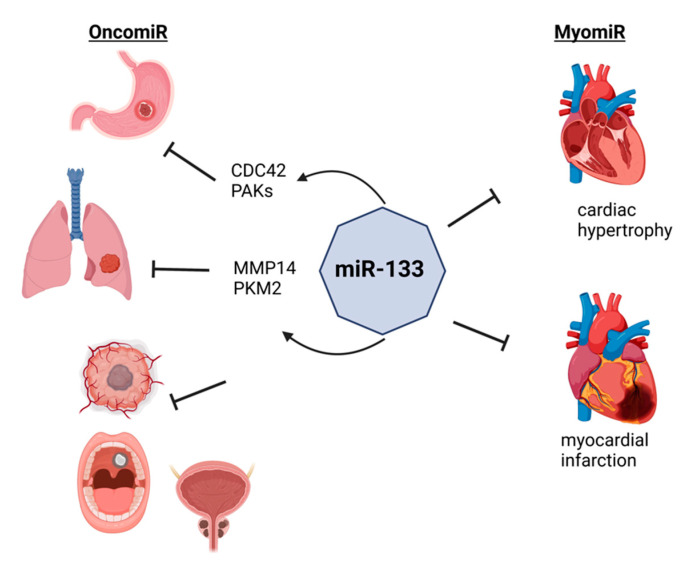
OncomiRs and myomiRs. MyomiRs have a dual function and are involved in tumorigenesis. *miR-133* regulates the cardiac hypertrophy and improves the myocardial function after infarction, while it is associated with multiple tumors. *miR-133* targets 3′-UTR of *CDC42* and regulates PAKs, thus preventing the growth and metastasis of gastric cancer. Similarly, *miR-133* prevents the proliferation, migration, and invasion of lung cancer by targeting *MMP14* and *PKM2*. Furthermore, *miR-133* is involved also in the pathogenesis of glioblastoma and oral and prostate cancer. Figure created with BioRender.com, accessed on 24 October 2022.

**Figure 2 cells-11-03551-f002:**
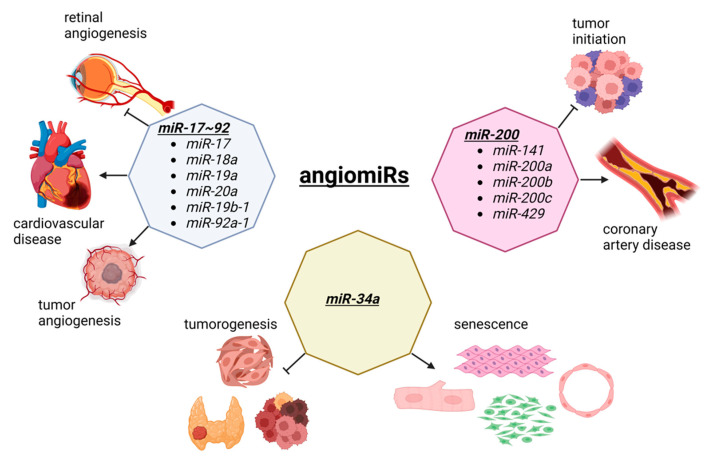
The dual role of angiomiRs in cancer and cardiac pathophysiology. The *miR-17~92* cluster is involved in tumorigenesis and tumor vascularization. This cluster is also involved in retinal angiogenesis and the progression of cardiovascular disease. The members of the *miR-200* family prevent the tumor initiation and malignant transformation, although they are upregulated in coronary artery disease. *MiR-34a* is a tumor suppressor involved in the development of thyroid cancer, head and neck squamous cell carcinoma, and cancer stem cells division. The overexpression of *miR-34a* suppress the proliferation and induces senescence in cardiomyocytes, fibroblasts, smooth muscle, and endothelial cells, by inhibiting sirtuin 1 (*SIRT1*). Figure created with BioRender.com, accessed on 24 October 2022.

**Figure 3 cells-11-03551-f003:**
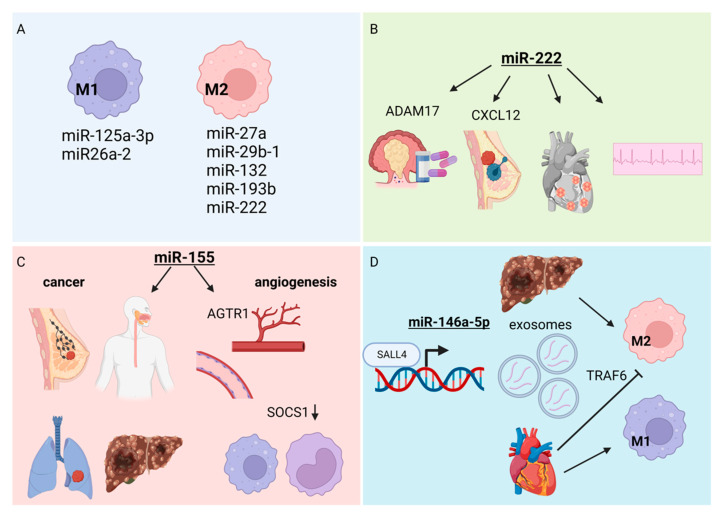
Immuno-miRs. (**A**) MiRNAs responsible for regulation of cytokine gene expressions leading to the differentiation of two types of monocytes-derived macrophages—inflammatory subtype M1 and anti-inflammatory subtype M2. (**B**) The role of *miR-222* in tumorigenesis and cardiovascular disease. *MiR-222* targets *ADAM17* to prevent multidrug resistant colorectal carcinoma. The inhibitory effect on the chemotaxis of tumor associated macrophages in breast cancer is mediated by targeting *CXCL12*. The overexpression of *miR-222* is associated with atrial fibrillation and Coxsackie virus caused myocarditis. (**C**) The overexpression of *miR-155* is associated lymph node metastasis in breast cancer and advance of esophageal, liver, and lung cancer. *Mir-155* regulates angiogenesis by controlling the expression of *AGTR1* in endothelial cells and *SOCS1* in monocytes/macrophages. (**D**) The extracellular RNA, *miR-146a-5p*, is highly presented in hepatocellular carcinoma derived exosomes and regulates the polarization of macrophages into M2 tumor-associated macrophages. On the contrary, the cardiomyocytes-derived *miR-146a-5p* inhibits the M2 macrophage polarization by targeting *TRAF6* while promoting M1 macrophage polarization. Figure created with BioRender.com, accessed on 24 October 2022.

**Figure 4 cells-11-03551-f004:**
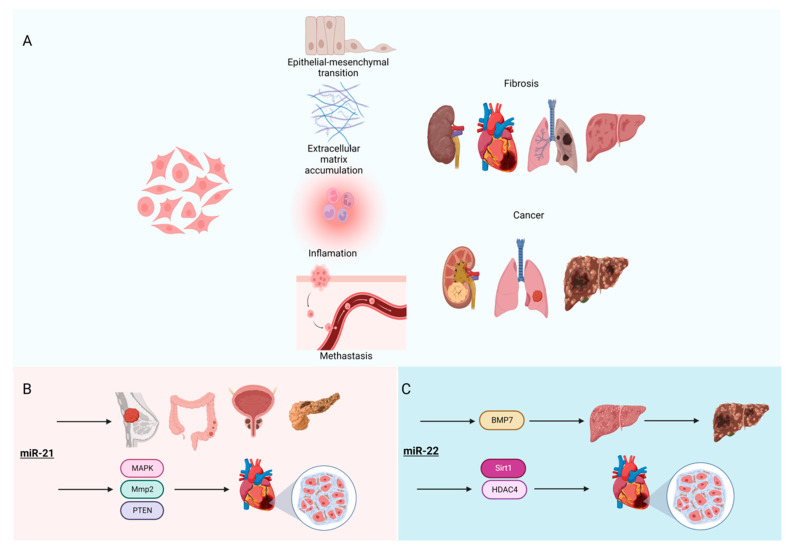
FibromiRs. (**A**) The core mechanisms of fibrosis and carcinogenesis. Multiple cell types (e.g., fibroblasts, myofibroblasts, epithelial cells, and macrophages) are involved. The pathophysiological mechanisms include inflammation, epithelial to mesenchymal transition, extracellular matrix accumulation, and metastasis. (**B**) *MiR-21* is a diagnostic biomarker for multiple cancers, including breast, pancreatic, colorectal, and prostate. *MiR-21* stimulates cardiac fibroblasts by targeting *Mmp2* and the PTEN pathway, leading to the progression of myocardial disease. (**C**) *MiR-22* induces the liver fibrosis through *BMP7* leading to progression into hepatocellular carcinoma. In the heart, *miR-22* promotes cardiac fibrosis by targeting *Sirt1* and *HDAC4*. Figure created with BioRender.com, accessed on 24 October 2022.

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
