# Peer review of "MicroRNAs in Cancer and Cardiovascular Disease"

_cells, 2022, doi:10.3390/cells11223551_

Round 1

Reviewer 1 Report

In the review article “MicroRNAs in Cancer and Cardiovascular Disease”The authors  reviewed the similarities between cancer and cardiovascular disease from the perspective of microRNAs (miRNAs). They explained how a specific set of miRNAs are both oncomiRs (miRNAs in cancer) and myomiRs (muscle-related miRNAs). Further miRNAs categories related to the similar pathogenesis angiomiRs, immunomiRs, and fibromiRs are discussed.

The introduction, figures, concluding remarks and references are framed correctly. Overall, the article is informative for the scientific fraternity. I recommend the paper for publication after rectifying the below mentioned minor errors.

Line No.115 Line correction required

Line No.169 Line correction required

Line No. 243 Line Correction required

In the reference, Page No. not given in the references 11 and 13.

Author Response

In the review article “MicroRNAs in Cancer and Cardiovascular Disease”The authors  reviewed the similarities between cancer and cardiovascular disease from the perspective of microRNAs (miRNAs). They explained how a specific set of miRNAs are both oncomiRs (miRNAs in cancer) and myomiRs (muscle-related miRNAs). Further miRNAs categories related to the similar pathogenesis angiomiRs, immunomiRs, and fibromiRs are discussed.

The introduction, figures, concluding remarks and references are framed correctly. Overall, the article is informative for the scientific fraternity. I recommend the paper for publication after rectifying the below mentioned minor errors.

Response: Thank you very much for your praise.

Line No.115 Line correction required

Line No.169 Line correction required

Line No. 243 Line Correction required

Response: The hyphenation of the word and line returns cannot be adjusted as the Word template from the journal is adjusted as such.

In the reference, Page No. not given in the references 11 and 13.

Response: The Endnote program was ran again to correct the errors in references.

Reviewer 2 Report

An interesting review the similarities between cancer and cardiovascular disease from the perspective of microRNAs (miRNAs). Some points should be noted as below.

1) Figure 1. OncomiRs and myomiRs. Besides miR-133, are there any other miRs related oncomiRs and myomiRs? IF there do, please add to Figure 1.

2)  Are there any miRs contribute to both the functional role of angiomiRs, immuno-miRs and fibromiRs? Please mention about it.

3) As to Cancer, there has been an increasing recognition that cancer should be best be conceptualized as a complex adaptive ecosystem (â‘ Willis A. The ecosystem: an evolving concept viewed historically. Funct Ecol 1997;11:268–271). An interesting paper recently showed that cancer should be a multidimensional spatiotemporal unity of ecology and evolution pathological ecosystem, in which cancer cells and the stromal cells transform, cooperate and even co-evolve with each other over time and space (â‘¡ Nasopharyngeal Carcinoma Ecology Theory: Cancer as Multidimensional Spatiotemporal Unity of Ecology and Evolution Pathological Ecosystem. Preprints. 2022; 2022100226. https://www.preprints.org/manuscript/202210.0226/v1). These views should be better complemented in the Introduction or Discussion section to make it more advanced.

Author Response

An interesting review the similarities between cancer and cardiovascular disease from the perspective of microRNAs (miRNAs). Some points should be noted as below.

Response: Thank you very much for your praise.

1) Figure 1. OncomiRs and myomiRs. Besides miR-133, are there any other miRs related oncomiRs and myomiRs? IF there do, please add to Figure 1.

Response: As stated in lines 76 - 78: "Compared to oncomiRs, the list of myomiRs is small, including miR-1, miR-133a/b, miR-206, miR-208a/b, miR-302, miR-367, miR-486, and miR-499 [46,47]. Not surprisingly, all myomiRs are involved in tumorigenesis." There are several myomiRs considered as oncomiRs. Of course, it is possible to give details about all myomiRs. Yet, such long list of cited publications may dilute the important message that we want to delivery by highlighting one or two miRNAs in each category as stated in the subsection of the Results section. To this end, we only list miR-133 as an example, instead of creating figures and images for each myomiRs.

2)  Are there any miRs contribute to both the functional role of angiomiRs, immuno-miRs and fibromiRs? Please mention about it.

Response: Yes, there are. The following sentence was added to the Conclusions section:

"To maintain the homeostasis of the tissues and remodelling of the tissues upon damages, angiogenesis, immune responses, and fibrosis are interconnected. As such, miRNAs are identified to be involved in each cellular activities as angiomiRs, immuno-miRs, and fibromiRs, respectively. Not surprisingly, some miRNAs (e.g., miR-17~92 cluster, miR-34a, and miR-200 family) are involved in all three cellular activities, which some overlapping miRNAs are responsible for such cellular activities and responses."

3) As to “Cancer”, there has been an increasing recognition that cancer should be best be conceptualized as a complex adaptive ecosystem (â‘ Willis A. The ecosystem: an evolving concept viewed historically. Funct Ecol 1997;11:268–271). An interesting paper recently showed that cancer should be a multidimensional spatiotemporal “unity of ecology and evolution” pathological ecosystem, in which cancer cells and the stromal cells transform, cooperate and even co-evolve with each other over time and space (â‘¡Nasopharyngeal Carcinoma Ecology Theory: Cancer as Multidimensional Spatiotemporal “Unity of Ecology and Evolution” Pathological Ecosystem. Preprints. 2022; 2022100226. https://www.preprints.org/manuscript/202210.0226/v1). These views should be better complemented in the Introduction or Discussion section to make it more advanced.

Response: Thank you very much for this valuable information and articles. We have added the following sentence to the Conclusions section:

“This is particularly interesting as cancer is considered as a complex adaptive ecosystem [157], in which cancer cells and the stromal cells transform, cooperate, and even co-evolve with each other over time and space [158]. Thus, it will be interesting to further investigate miRNAs from the perspective of ecosystem in cancer and possibly in cardiovascular disease.”